# New Approach of Metals Removal from Acid Mine Drainage

**Radmila Markovic** [1,*] **, Masahiko Bessho** [2,3] **, Nobuyuki Masuda** [3,4] **, Zoran Stevanovic** [1] **, Dragana Bozic** [1] **, Tatjana Apostolovski Trujic** [1] **and Vojka Gardic** [1]

1 Mining and Metallurgy Institute Bor, Zeleni bulevar 35, 19210 Bor, Serbia; zoran.stevanovic@irmbor.co.rs (Z.S.); dragana.bozic@irmbor.co.rs (D.B.); tatjana.trujic@irmbor.co.rs (T.A.T.); vojka.gardic@irmbor.co.rs (V.G.)

2 International Center for Research and Education on Mineral Engineering and Resources, Akita University, 1–1 Tegata Gakuen-machi, Akita 010–8502, Japan; bessho@gipc.akita-u.ac.jp

3 Graduate School of International Resource Sciences, Akita University, 1–1 Tegata Gakuen-machi, Akita 010–8502, Japan; masudanobuyuki@gmail.com

4 Mitsui Mineral Development Engineering Co., Ltd., 1–11-1 Osaki, Shinagawa-ku, Tokyo 141–0032, Japan

* Correspondence: radmila.markovic@irmbor.co.rs

**Abstract:** The possibility of metal removal from the real acid mine drainage (AMD) in the area of copper ore mines in the southeast Serbia, was investigated through a combination of neutralization and adsorption methods. This approach of metal removal from AMD includes a two-step neutralization method in the first phase, aiming to separate metals as sludge. The results of laboratory test revealed that more than 99 mass % of Fe is removed up to pH 4 and more than 99 mass % of Cu up to pH 7. Based on the results obtained in laboratory conditions, a test on a semi industrial plant was carried out. The two-step neutralization separately removed Fe and Cu at pH 4 and 7, respectively. Especially, the obtained sludge at pH 7 included 1.24 mass % of Cu, much higher than usual Cu ore. Chitosan was applied for dissolved Mn removal from treated AMD. After 24 h incubation, 70 mass % of Mn is removed from the treated AMD at pH 7.4. Mn concentration was reduced from approx. 35 mg $L^{-1}$ to 5 mg $L^{-1}$. These results have indicated that a combination of neutralization and adsorption methods could be used effectively for metal removal from real AMD.

**Keywords:** acid mine drainage; metals removal; neutralization; adsorption; chitosan; semi industrial plant

## 1. Introduction

Natural resources such as coal, iron, nonferrous metals, precious metals, industrial minerals, etc., are of integral importance for the development of human societies. During the World Summit on Sustainable Development (Johannesburg, South Africa, 2002), it was decided that a concept of sustainable development would be based on the environmental protection as one of the three main pillars, both for the developed and developing countries. Mining and mineral processing on the Balkan Peninsula, including Albania, Bosnia and Herzegovina, North Macedonia, Kosovo (Territory under the Interim UN Administration), Montenegro, and Serbia, has played a vital role in the European history and economy. Actually, this area was a major part of the supplies of copper, lead, and zinc in Europe until 1990 [1]. The global mining activities (mineral and metal production processes) produce several billion tons of solid inorganic wastes or by-products, including the liquid waste [2]. According to the study made for the Environment Directorate-General, European Commission, more than 4.7 billion tons of mining waste and 1.2 billion tons of tailings waste are stored all over the European Union [3].

The open pit or underground mining operations generally have a serious negative impact on the surrounding environment, such as the air pollution, land use and biodiversity, and water availability. Noise and vibrations, the use of energy sources, and visual effects are sometimes negative consequences of the mining and metallurgical activities. In addition to the active sites, thousands of the "old" or "depleted" sites are scattered in the region. Furthermore, some effluents generated in the metal mining industry contain large quantities of toxic substances (cyanides, heavy metals, and other harmful and dangerous substances), which have serious human health and ecological implications [4–6].

Mine water generated from the active as well as the abandoned mines is one of the main chemical threats to the groundwater and surface water quality. Mine waters are, as a rule, acidic with pH value mostly between 2.5 and 4 due to an elevated concentration of sulfuric acid, as the second product of bacterial oxidation of sulfide minerals. Pyrite is the most abundant mineral in the poly-metallic sulfide ore deposits and mining waste dumps. The oxidation of pyrite and copper minerals in an aqueous environment occurring via two simultaneous mechanisms, i.e., biochemical involving bacteria, and chemical way, can be described by the following stoichiometric reactions [7–9]:

$$\text{(bacterial)} \quad 2FeS_2 \, + \, 7.5O_2 \, + \, H_2O \rightarrow Fe_2(SO_4)_3 \, + \, H_2SO_4 \tag{1}$$

$$\text{(chemical)} \quad 2FeS_2 \, + \, 7Fe_2(SO_4)_3 \, + \, 8H_2O \rightarrow 15FeSO_4 \, + \, 8H_2SO_4 \tag{2}$$

$$\text{(chemical)} \quad FeS_2 \, + \, Fe_2(SO_4)_3 \rightarrow 3FeSO_4 \, + \, 2S^0 \tag{3}$$

$$\text{(bacterial)} \quad S^0 \, + \, H_2O \, + \, 1.5O_2 \rightarrow H_2SO_4 \tag{4}$$

Generation and release of acid mine drainage (AMD) containing elevated concentrations of metals from the mine waste induces an environmental problem on a global scale, and some kind of method, active or passive treatment, is required for the mine drainage [10]. The active treatment methods use neutralizing agents to raise the pH up to the effluent water quality standards [11–15]. Mine water originating from the active or closed copper mines contains copper ions sometimes in a considerable concentration usually associated with an equivalent or even two times higher concentration of $Fe^{2+}/Fe^{3+}$ ions as a consequence of bacterial leaching the sulfide copper and iron minerals. The presence of other heavy metal ions (Mn, Cd, Zn, Pb, Ni, etc.) in the mine water depends on mineralization of an ore body, but their concentration is much lower than the concentration of copper or iron. The most environmentally effective techniques available for the AMD treatment are the internal neutralization methods, water-covers and biological/natural degradation processes. A number of factors dictate the level of the treatment system that is necessary to ensure that the effluent standards will be met. These include the chemical characteristics of the AMD; quantity of water in need of treatment; local climate; and sludge characteristics. The chemicals usually used for the AMD treatment include limestone, hydrated lime, soda ash, caustic soda, carbide lime, ammonia, calcium peroxide, and fly ash.

Slaked lime neutralization, referred to in the mining circles as the "chemical process", is often used to treat the AMD in the mining industry. Previous studies showed that slaked lime can treat the wastewater effectively. It had the capability to increase pH value and also can treat heavy metals for example at pH ranges 10–11 for zinc, 9.2–11.6 for lead, 4–11.8 for iron, and 7–11.8 for copper [16]. Another study showed that slaked lime can be used to treat an effluent from a dairy farm using the coagulant techniques and the results indicated that the suspended solids, organic matter, and nitrogenous and phosphate compounds were reduced [17].

The neutralization method can remove the dissolved metals from the wastewater due to the pH dependence of the metal solubility, and it is a very effective operation for wastewater purification. However, for some metal ions, such as Ni and Mn, a higher pH value over the effluent standard is required to remove them as a sludge precipitation. In this case, more operation is needed to lower the pH value below the effluent standard. If the processing in the pH region above the effluent standard needs to be avoided, other removal methods must be applied for that.

Adsorption is one of the effective techniques for removal the metal ions from water. Application of organic or inorganic adsorbents to wastewater treatment has been reported [18–20]. Especially, the natural materials that are available in large quantities, or certain waste products from the industrial or agricultural operations, may have a potential as the inexpensive sorbents for heavy metals from AMD. Due to their low cost, after these materials have been expended, they can be disposed of without expensive regeneration [18]. Most of structural proteins are found in the skin, tendons, cartilage, bones, and connective tissues of animals. In the papers [21,22] are presented very useful data about the main sources, properties, and chitin application as well as properties and application of chitosan, the most important derivative of chitin. Chitosan is a cationic polymer obtained from the chitin comprising copolymers of β (1 → 4)-glucosamine and N-acetyl-D-glucosamine [23]. Chitosan is soluble in the acidic aqueous media, and is used in many applications (food, cosmetics, biomedical, and pharmaceutical applications). Furthermore, chitosan has applications in drinking water and wastewater treatment due to its ability to remove metallic ions from solutions. The ability of chitosan to remove arsenic from natural water, and copper and zinc from mining wastewater was verified [19]. This polymer has been shown to be the biologically renewable, biodegradable, biocompatible, non-antigenic, non-toxic, and biofunctional. Metal removal caused by the adsorption is usually affected by the electrostatic interaction, chelating effect or chemical bonds. Thus, it is expected that the adsorption method can remove metals which have a higher solubility in an acidic or neutral range below the effluent standard pH value.

The aim of the present investigation was to examine the possibility of purification of the AMD generated from the copper ore mining activities in southeast Serbia by combining the neutralization and adsorption methods. During more than one hundred years of copper ore mining activities in southeast Serbia, the mine drainage water has been released to the downstream without any treatment through tributaries of the Danube River Basin and has provided the negative environmental impact on the river water of the Danube River. The results of research realized in the period from 2011 to 2013 and financed by the Japan International Cooperation Agency (JICA) and Japan Society for the Promotion of Science (JSPS) have shown that the environmental effect of AMD, generated in the selected area on the water in the Danube River, is not so clear. However, the strongly serious environmental impacts are detected on 30 km along the Bela River which is a tributary of the Timok River which flows to the Danube River. The generation of AMD from weathering of sulfidic ore and disposal overburden in the copper ore mine treatment plant in southeast Serbia continues to significantly impact local and regional water resources (Figure 1).

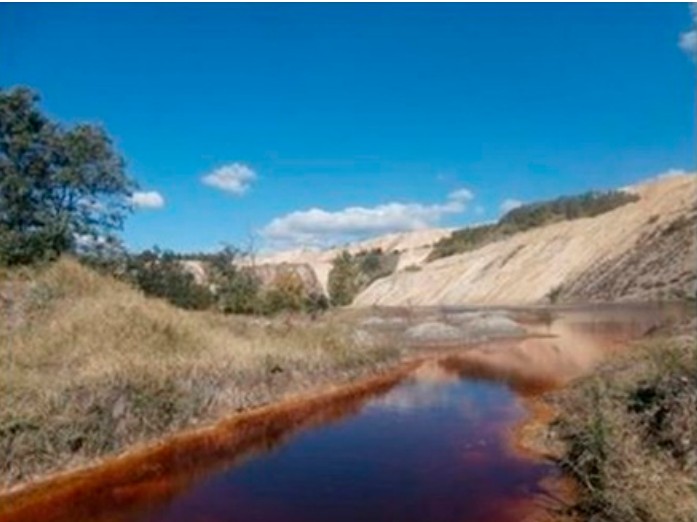

**Figure 1.** The Robule Lake—example of acid mine drainage (AMD) in the area of the copper ore mine treatment plant in the southeast of Serbia.

This investigation includes not only purification of the AMD but also recovery of some valuable metals which dissolve in the AMD as a resource. The conventional neutralization treatment prefers to remove dissolved metals for AMD purification. Thus, metal sludge generated by the neutralization process is generally stored in disposal sites as industrial wastes. However, the AMD usually contains some valuable metals. Recovering these metals positively will lead, not only to the efficient use of metal resources, but also reduction of waste metal sludge. Finally, it may contribute to prolongation of lifetime disposal sites.

The European and local legislation rules and concerns for environmental protection requests the removal of heavy metal ions from wastewater, in order to keep their concentrations below the maximum allowable value required by the law [24], prior to their releasing into the environment. In this study, a two-step neutralization method by controlling pH value was firstly applied to separate metals in sludge during each neutralization step. Precipitation efficiency of heavy metal ions during the neutralization process on different pH values was mainly investigated in the laboratory conditions. Investigations of the two-step neutralization process are continuing on a semi industrial plant that was designed and constructed based on the results of laboratory tests. Then, an adsorption method was mainly applied for $Mn^{2+}$ ions removal. The mine wastewater often includes the Mn ion. Because the pH value for the Mn ion precipitation is sometimes much higher than the national effluent standard, the neutralization method requires a higher pH (more than 8) to remove Mn ions from the wastewater. For that, the other approach is needed to prevent the neutralization method at a higher pH value (more than 9–10). In order to investigate the possibility for a new approach of metal ions removal, some organic materials were additionally applied to the treated AMD neutralized through the semi industrial plant at around pH 7–8 below the effluent standard.

## 2. Materials and Methods

### 2.1. Materials

#### 2.1.1. AMD Sample

Experimental procedure was carried out with a real sample of AMD from location Robule Lake (Figure 1), generated in the area of copper mining activities in the southeast of Serbia. The Robule Lake was formed by seepage and underground water from the waste dump site consisting of a mine waste material mixture (waste rocks, flotation tailings, overburden, and low-grade copper ores). The mixed mine waste material contains the oxidized copper minerals as well as pyrite [7]. It suggests that the seepage water from the dump contains copper.

Monitoring of the chemical characteristics of water from the Robule Lake was realized quarterly in the period from January 2017 to December 2017. The results for major elements (Table 1) revealed that this water had the typical characteristics of AMD, and this location was chosen for the planed investigations.

**Table 1.** Chemical characterization of the Robule Lake from January 2017 to December 2017.

| Month/Year | pH | Element | | | | | | | |
|---|---|---|---|---|---|---|---|---|---|
| | | Al | Fe | Mg | Mn | S | Zn | Cu | Sr |
| | | Concentration, mg L$^{-1}$ | | | | | | | |
| March/2017 | 2.96 | 299.6 | 613.2 | 1279.6 | 91.8 | 3366 | 23.1 | 46.5 | 0.8 |
| June/2017 | 2.94 | 294.7 | 506.9 | 1154.2 | 103.4 | 3203 | 22.6 | 38.3 | 1.3 |
| September/2017 | 2.81 | 209.8 | 322.6 | 893.5 | 90.8 | 2377 | 12.8 | 34.7 | 1.2 |
| December/2017 | 2.80 | 261.5 | 456.3 | 979.1 | 87.3 | 2753 | 19.2 | 40.7 | 1.3 |

### 2.1.2. Slaked Lime

Content of some minerals in the slaked lime sample that is used as a neutralizer was specified by the X-ray diffraction using a GNR Explorer diffractometer under the following conditions: Cu Kα at wavelength 1.54 Å; voltage U = 40 kV; current I = 30 mA; detector: scintillation counter; geometry of the apparatus: θ-θ. The identified minerals are as follows: 60.3 mass % of $Ca(OH)_2$ and 39.7 mass % of $CaCO_3$. Slaked lime is a local available.

### 2.1.3. Flocculant

Flocculant ACCOFLOC/ARONFLOC, brand A-95 (purchased from MT Aqua Polymer, Inc., Japan) was used in the process of suspended particles settling after different neutralization steps. Flocculant A-95 is polyacrylamide, week anion, molecular weight is 1700, proposed for the solution pH a range from 6–8.

### 2.1.4. Adsorbent

As an adsorbent material for metal removal, chitosan with medium molecular weight was purchased from Sigma-Aldrich Japan Co., Japan. This chitosan sample had a deacetylation degree of 75% to 85% and 1 wt% of the chitosan in 1% acetic acid at 25 °C had 200–800 cps of viscosity. The chitosan sample having a shape of powder and/or chips was used without any purification for the adsorption removal test.

### 2.2. Neutralization Method

### 2.2.1. Laboratory Tests

Investigations were carried out with 1000 mL of real AMD samples. Lime milk, concentration of 2.5 mass % was used as a neutralizer to reached pH 3, 3.5, and 4 in a first neutralization step. The certain amount of lime milk was continually added to the AMD sample in a batch reactor. Stirring was performed with a magnetic stirrer, at a constant speed of 400 rpm, in order to avoid precipitation. After solution reached pH of 3, 3.5, or 4, each solution was filtered by the vacuum filtration to separate the solid from the liquid phase. Dewatered sludge was dried to a constant mass at 40 °C. The liquid sample was collected and given on chemical analyses according to the chemical characterization of the real sample. AMD sample neutralized in the first neutralization step up to pH 3, 3.5, or 4 was used as the start sample for the second neutralization step. Lime milk (2.5 mass %) was used as a neutralizer to reach pH of 7, 7.5, and 8. As in the first neutralization step, liquid and solid phases were separated by the vacuum filtration. The liquid phase was used for the chemical analyses and solid phase was dried at 40 °C and measured after reaching the constant mass value.

### 2.2.2. Two-Step Neutralization Treatment on Semi Industrial Plant

Different results of tests carried out in the laboratory condition indicated that the pH control process is effective for separate metal removal. All data obtained during the investigation of artificial and real wastewater from defined location in the area of copper mining activities in the southeast of Serbia were used for the design of a new semi industrial plant for continuous neutralization process. The new semi industrial plant was designed and manufactured in Japan, by Mitsui Mineral Development Engineering Co., during the activities of SATREPS project: "Research on the integration system of spatial environment analyses and advanced metal recovery to ensure sustainable resource development" from 2015 to 2019, which has been conducted by the Akita University, Japan Space System, Mitsui Mineral Development Engineering Co., Ltd., and the Mining and Metallurgy Institute Bor. The new plant was installed in the Mining and Metallurgy Institute Bor. Designed process capacity of the equipment is 2–7 L min$^{-1}$ and it was capable to use in the field continuous operation.

### 2.2.3. General Description of the Semi Industrial Plant for Two-Step Neutralization Treatment

Specification of the semi industrial plant (Figure 2), which consists of five Units, a Compressor, and a Control Panel, and that continuously performs the pH control and flocculation-precipitation of the AMD in two steps, is given in Table 2.

**Table 2.** General description of the semi industrial plant for two-step neutralization treatment.

| Items | Description |
|---|---|
| Unit A: Wastewater supply | Receiving pump, flow rate: 20 L min$^{-1}$, used for transport the feed wastewater in the wastewater tank. Wastewater tank, for storage of feed wastewater. Wastewater pump, flow rate: 2–7 L min$^{-1}$, used for feed wastewater transport in unit B. |
| Unit B: pH control and flocculation-precipitation, configured from: Unit B-1, Unit B-2 and Unit B-3 | Unit B-1: Thickener A, volume 410 L, used for solid–liquid separation of the feed water that has been done primary pH control (according to the first set pH value) and flocculation. Unit B-2: pH Control tank A/B (50 L each tank), Flocculation tank A/B, (50 L each tank). This unit is used for the primary pH control (according to the first set pH value), secondary pH control (according to the second set pH value) and flocculation of the feed wastewater. A pH control agitator A/B is used for solution mixing in the pH Control tank A/B. Flocculation agitator A/B is used for solution mixing in the Flocculation tanks. Unit B-3: Thickener B, volume 410 L, used for the solid-liquid separation of the feed water that has been done the secondary pH control (according to the second set pH value) and flocculation. |
| Unit E: Filtration | Thickener A and Thickener B of Unit B should be visually checked regularly for slurry buildup, and as it has grown to a substantial level, manually operated valves under the respective thickeners should be opened for discharge of slurry by a slurry pump, flow rate 20 L min$^{-1}$, to Filter press, capacity 3 L. The filtrate will be discharged by a portable filtrate pump or returned to the wastewater tank depending on the level of its water quality. Filter press will be of a fully automatic type with a pneumatic press function built in. |
| Unit C: Slaked lime feed | Slaked lime and water are to be charged into a Lime slurry tank, volume 250 L, in advance for controlling its concentration to 2.5 mass %. The conditioned lime sludge will be sent automatically, by lime sludge pump A/B, (0.5 L min$^{-1}$) to pH Control tank A/B according to their respective pH values. |
| Unit D: Flocculant feed | Polymer flocculant and water are charged into the Flocculant tank, (volume 100 L), in advance for controlling its concentration up to 0.5 g L$^{-1}$. The conditioned flocculant will be sent to the Flocculation tank A/B at a constant feed rate irrespectively of the feed water flow rate by a flocculant pump A/B, flow rate 0.05 L min$^{-1}$. |
| Compressor | Compressor (370 L min$^{-1}$) will be used in order to supply the compressed air to filter press and solenoid valves. |
| Control panel | Control panel will be used for control, operation and power supply of equipment |

Storage tank for wastewater (volume 1000 L) is used for storage for the sampled AMD from the proposed location. Ambient conditions are 5–40 °C; Power Supply: 220 VAC × 3-phase × 50Hz, 13 kVA. Equipment is made from acid resistance material due to the fact that the acidity of AMD is mostly between 2.5 and 4, that is some as for pH of AMD from the Robule Lake (Table 1).

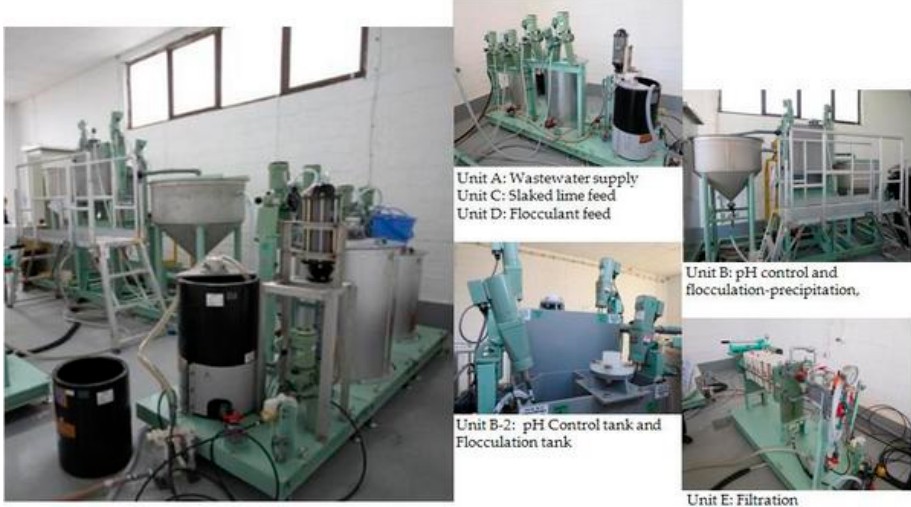

**Figure 2.** Semi industrial plant for two-step neutralization treatment.

2.2.4. Technological Procedure for Neutralization Treatment on Semi Industrial Plant

A sample from the Robule Lake was used for the test operation. The AMD was fed to the equipment and pH was set; for the first step on pH 4 and for the second step on pH 7. The wastewater from storage tank for wastewater was transported by a receiving pump with constant flow rate of 20 L min$^{-1}$ to the wastewater tank. The wastewater tank was with a level control that switches off/on the receiving pump. The wastewater pump with constant flow rate of 5 L min$^{-1}$ was used for wastewater transport to the pH Control tank A. At the same time, to the pH Control tank A was continually added lime milk (2.5 mass %) from the Lime slurry tank by a lime slurry pump. In the pH Control tank A, is installed a pH meter aimed at controlling the first set pH value. Due to the fact that the primary pH value was set on value of 4, according to the pH value during the test, dosing of lime milk was corrected. Suspension from the pH Control tank A overflowed in the Flocculation tank A. Flocculant (0.05 mass %) was transported by a flocculant pump A with a constant flow rate of 0.025 L min$^{-1}$ to the Flocculation tank A. Sludge was precipitated in the bottom of Thickener A incrementally during the test. Neutralized AMD that overflowed from the Thickener A was taken for a chemical analysis and by the gravity flow to the pH Control tank B for the secondary pH control (according to the second set pH value of 7). Flocculation was followed in the Flocculation tank B thereafter, and then a solid-liquid separation occurred in the Thickener B. Neutralized AMD that overflowed from the Thickener B was also taken for the chemical analysis and discharged continuously by the gravity as the treated water. Thickener A and Thickener B were discharged one by one, aiming to filtrate the slurry pulp on a filter press. The filtrate was discharged by a portable filtrate pump as the treated water. Dewatered sludge was measured and dried at 40 °C to constant mass and taken for a chemical analysis.

*2.3. Adsorption Method*

Adsorption removal tests for Mn$^{2+}$ ions removal using chitosan as an adsorbent were mainly performed at room temperature. Basic tests were firstly carried out to confirm the Mn$^{2+}$ ions' adsorption capacity. For that, 0.5 mM of Mn solutions with various pH were prepared from manganese sulfate pentahydrate (MnSO$_4$·5H$_2$O), which was purchased from Wako Pure Chemical Industries, Ltd., Japan. The solution pH was adjusted using the NaOH aqueous solution. The desired amount of chitosan (0.5–5 g) was introduced into 50 mL of Mn solution. The Mn solutions including chitosan were stirred with a magnet bar for 24 h. After the adsorption test, the solution samples were filtrated through a PTFE membrane filter with a 0.45 μm mesh for a quantitative analysis.

As for application of the adsorption method to the AMD treatment, the real wastewater treated by the neutralization process was used. To the real AMD water derived from the Robule Lake, the two-step neutralization including treatment at pH 4 and 7 was applied. The treated AMD water had the value of pH 7.2 and included 34 mg $L^{-1}$ of $Mn^{2+}$ ions. Chitosan (5 g) was introduced into 50 mL of the treated AMD water. It was stirred with a magnet bar for 24 h. After an adsorption removal test, the treated AMD water was filtrated through a PTFE membrane filter with a 0.45 μm mesh for a quantitative analysis.

### 2.4. Quantitative Analysis of Elements

For a quantitative analysis, the metal element concentrations in the test samples and chemical composition of sludge were mainly measured by atomic emission spectrometry with inductively-coupled plasma technique (AES-ICP, Spectro Ciros Vision) and inductively coupled plasma mass spectrometry (ICP-MS, Agilent 7700). Chemical analyses were done in duplicate with accompanied quality control (blank and certified reference materials (CRM) analysis). Based on the results of these analyses, the metal removal rate was also calculated.

## 3. Results and Discussion

### 3.1. Neutralisation Method

#### 3.1.1. Laboratory Tests

Chemical composition of AMD from the Robule Lake, (mg $L^{-1}$): Al—209.8; Fe—322.6; Mg—893.1; Mn—90.8; Zn—12.8; Cu—34.7; Co—0.87; S—2376.2; Sr—1.7; Ni—0.413; As—0.004; Se—0.008 and Cd—0.039. Site measurement indicated: pH—2.87; ORP—793 mV and flow rate—3489 L $min^{-1}$. Concentration of Pb, Cr and Cs ions were below the sensitivity limit of used analytical method, (μg $L^{-1}$): Pb < 0.1; Cr < 0.5 and Cs < 0.1.

The main point of AMD purification by the neutralization method using lime milk is the conversion of soluble metal forms into insoluble ones. The increased pH allows reactions between the metal ions present in the wastewater and hydroxide ions to form the following precipitates: $Al(OH)_3$; $Fe(OH)_3$; $Co(OH)_2$; $Ni(OH)_2$; $Cu(OH)_2$; $Pb(OH)_2$; $Fe(OH)_2$; $Zn(OH)_2$; $Mg(OH)_2$, etc.

Table 3 shows the results of neutralization tests during the first neutralization step on different pH values: 3, 3.5 and 4.

**Table 3.** Concentration of some elements in AMD after the first neutralization step.

| Neutralization from pH Start Value of 2.78 | Element | | | | | | | Neutralizer |
|---|---|---|---|---|---|---|---|---|
| | Fe | Mn | Cu | Zn | Cd | Co | Ni | 2.5 Mass % Lime Milk |
| | Concentration, mg $L^{-1}$ | | | | | | | Consumption, mL $L^{-1}$ |
| start | 322.6 | 90.8 | 34.7 | 12.8 | 0.040 | 0.87 | 0.41 | - |
| to pH 3 | 274.2 | 62.9 | 31.6 | 12.2 | 0.041 | 0.86 | 0.42 | 3.5 |
| to pH 3.5 | 12.3 | 62.9 | 31.2 | 12.1 | 0.041 | 0.87 | 0.42 | 20.8 |
| to pH 4 | 1.0 | 62.7 | 31.5 | 12.0 | 0.041 | 0.86 | 0.42 | 25.0 |

Based on the results for each element starting concentration and concentration after neutralization at different pH values (Table 3), a removal degree for each element was calculated. At pH 3.5 and pH 4 Fe removal degree had the values of 96.20 mass % and 99.70 mass %, respectively, while about 15 mass % of Fe was removed at pH 3. Those data revealed the maximum Fe conversion into the insoluble hydroxide form occurred at pH 4. Thus, it was confirmed that Fe could possibly be separated from AMD at pH 4 during the first neutralization step. Cu removal degree at different pH values is almost constant (8.99 mass %, 10.06 mass %, and 9.37 mass % for pH 3, 3.5, and 4, respectively).

Figure 3 shows a dependence of pH on Zn, Cd, and Co removal degree. Mn concentration was reduced during the neutralization at different pH values, and removal degree was about 30 mass % in any case.

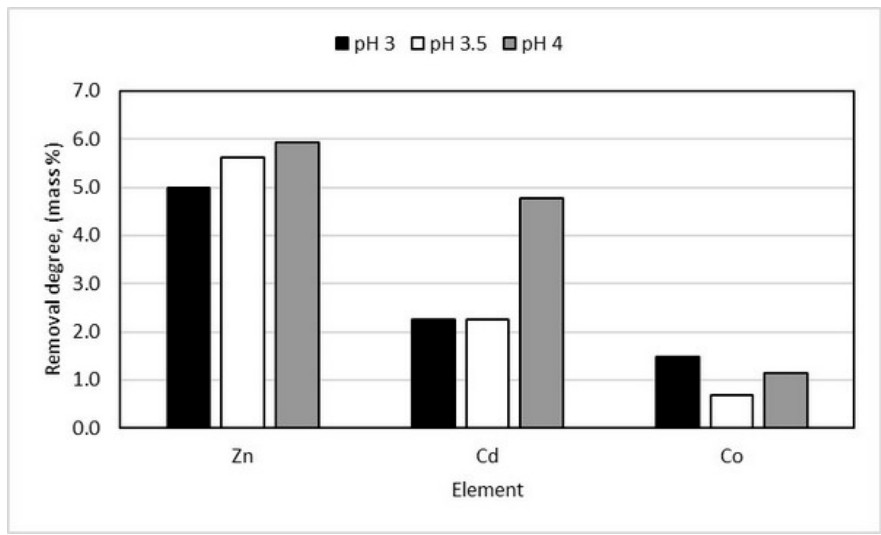

**Figure 3.** Dependence of Zn, Cd, and Co removal degree of pH value.

It was confirmed that a removal degree of Zn, Cd, and Co was between 1 and 6 mass % regardless of pH. These elements are originally minor component and generally have a higher solubility at around pH 4. Thus, it is considered that removal of these elements was caused by the co-precipitation with Fe hydroxides.

Ni concentration was not changed during the neutralization on pH up to 4 (Table 3). Lime milk consumption is lower for 17 mass % during the AMD neutralization up to pH 3.5 in comparison with lime milk consumption during the AMD neutralization up to pH 4 (Table 3).

Concentration of some elements after the second neutralization step of real AMD is presented in Table 4.

**Table 4.** Concentration of some elements after the second neutralization step of real AMD.

| Neutralization from pH 3 | Element | | | | | | | Neutralizer |
|---|---|---|---|---|---|---|---|---|
| | Fe | Mn | Cu | Zn | Cd | Co | Ni | 2.5 Mass % Lime Milk |
| | Concentration, mg L$^{-1}$ | | | | | | | Consumption, mL L$^{-1}$ |
| to pH 7 | 0.14 | 38.2 | 0.06 | 0.44 | 0.012 | 0.28 | 0.15 | 67.2 |
| to pH 7.5 | 0.03 | 28.4 | 0.04 | 0.13 | 0.001 | 0.13 | 0.08 | 70.0 |
| to pH 8 | 0.09 | 20.0 | 0.03 | 0.03 | 0.002 | 0.03 | 0.04 | 75.4 |
| **Neutralization from pH 3.5** | **Element** | | | | | | | **Neutralizer** |
| | Fe | Mn | Cu | Zn | Cd | Co | Ni | 2.5 mass % lime milk |
| | Concentration, mg L$^{-1}$ | | | | | | | Consumption, mL L$^{-1}$ |
| to pH 7 | 0.02 | 43.0 | 0.06 | 1.04 | 0.022 | 0.41 | 0.24 | 37 |
| to pH 7.5 | 0.02 | 37.2 | 0.03 | 0.27 | 0.013 | 0.20 | 0.12 | 40 |
| to pH 8 | 0.02 | 31.9 | 0.03 | 0.07 | 0.007 | 0.09 | 0.07 | 43 |
| **Neutralization from pH 4** | **Element** | | | | | | | **Neutralizer** |
| | Fe | Mn | Cu | Zn | Cd | Co | Ni | 2.5 mass % lime milk |
| | Concentration, mg L-1 | | | | | | | Consumption, mL L$^{-1}$ |
| to pH 7 | 0.01 | 42.2 | 0.04 | 0.65 | 0.019 | 0.33 | 0.21 | 31.8 |
| to pH 7.5 | 0.01 | 39.8 | 0.05 | 0.22 | 0.015 | 0.23 | 0.14 | 33.8 |
| to pH 8 | 0.01 | 36.8 | 0.03 | 0.11 | 0.010 | 0.16 | 0.10 | 40.3 |

Copper concentration was drastically reduced during the neutralization on different pH value (7, 7.5, and 8) below the value of maximum allowed concentration of 1 mg L$^{-1}$ [24]. At each pH, Mn concentration was much higher than the maximum allowed value (1 mg L$^{-1}$) (Table 4).

Figure 4 shows the total lime consumption at two different pH value during two neutralization steps of real AMD from the Robule Lake.

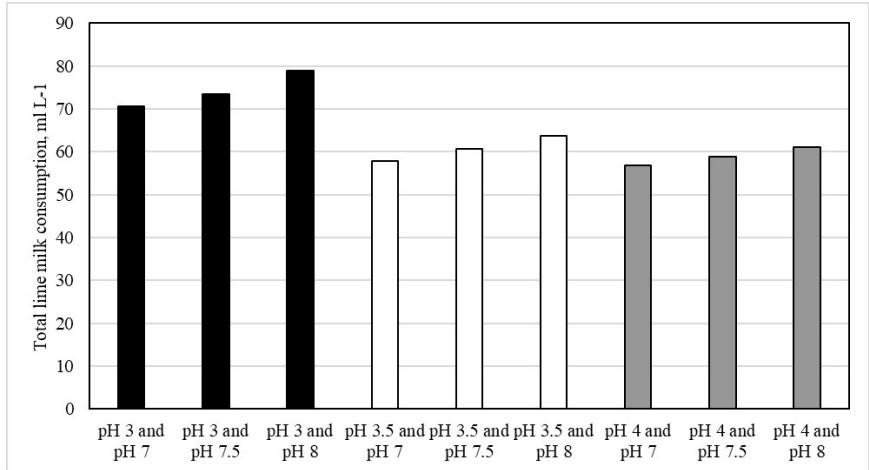

**Figure 4.** Total lime milk consumption during two neutralization steps of real AMD from the Robule Lake.

The total lime milk consumption had the minimum value for the first neutralization step on pH 4 and second neutralization step on pH 7 (Figure 4). Based on the above discussion about the concentration of some elements after two neutralization steps and the value of the total lime consumption at different pH values, a test on a semi industrial plant was carried out as the following: neutralization up to pH 4 as the first neutralization step and neutralization up to pH 7 as the second neutralization step.

### 3.1.2. Two-Step Neutralization Test on Semi Industrial Plant

The real AMD sample (about 1800 L) from the Robule Lake was used for the test operation. Characteristics of AMD that were sampled from a flow of the Robule Lake are given in Table 5. The AMD was sampled by a pump from the Robule Lake in two IBC containers (1000 L volume per one) and transported by a track to the laboratory in the Mining and Metallurgy Institute Bor where semi industrial plant for the two-step neutralization treatment is located.

During the two-step neutralization treatment, the following parameters were measured and controlled: pH value in the Tanks for pH control, AMD flow rate, lime milk, and flocculant flow rates. Process duration was 6 h.

Water samples for chemical analysis were taken from the Thickener A on each hour during the test. Sampling from the Thickener B started 3 h after starting the outflow of AMD neutralized in the second step. The results for chemical composition of the neutralized AMD during the two-step neutralization treatment are presented in Table 5.

**Table 5.** Concentration of elements in AMD neutralized in the two-step neutralization treatment on the semi industrial plant.

| Sample Location (Test Duration) | Element | | | | | | | |
|---|---|---|---|---|---|---|---|---|
| | Fe | Mn | Cu | Zn | Cd | Co | Ni | Ca |
| | Concentration, mg L$^{-1}$ | | | | | | | |
| Robule Lake (0 h) | 461.8 | 83.6 | 45.3 | 16.9 | 0.055 | 1.05 | 0.52 | 453 |
| First neutralization step, pH 4 | | | | | | | | |
| Thickener A (1 h) | 16.0 | 66.7 | 35.9 | 15.0 | 0.050 | 0.92 | 0.44 | 797 |
| Thickener A (2 h) | 11.5 | 65.3 | 36.6 | 14.7 | 0.050 | 0.94 | 0.46 | 861 |
| Thickener A (3 h) | 12.5 | 60.5 | 38.1 | 15.0 | 0.051 | 0.97 | 0.47 | 772 |
| Thickener A (4 h) | 7.5 | 64.4 | 37.0 | 15.2 | 0.052 | 0.97 | 0.46 | 783 |
| Thickener A (5 h) | 0.7 | 79.7 | 45.4 | 15.6 | 0.054 | 0.99 | 0.49 | 516 |
| Thickener A (6 h) | 8.3 | 74.2 | 40.3 | 15.0 | 0.050 | 0.91 | 0.43 | 860 |
| Second neutralization step, pH 7 | | | | | | | | |
| Thickener B (3 h) | 0.078 | 25.9 | 0.38 | 0.43 | 0.011 | 0.11 | 0.047 | 1430 |
| Thickener B (4 h) | 0.016 | 25.9 | 0.20 | 0.38 | 0.011 | 0.12 | 0.053 | 1107 |
| Thickener B (5 h) | 0.017 | 27.9 | 0.20 | 0.36 | 0.011 | 0.13 | 0.064 | 1298 |
| Thickener B (6 h) | 0.044 | 29.4 | 0.31 | 0.45 | 0.012 | 0.13 | 0.079 | 1312 |

The obtained results revealed that concentration of all elements, except for calcium, decreased. Increase in Ca concentration was due to the input of lime milk used as a neutralizer. Fe concentration was reduced during the first neutralization step below the maximum allowable value [24]. During the first neutralization step for a test duration of 6 h, the Fe (ferric iron) concentration was gradually reduced. Fe removal degree was in the range of 96.5 mass % to 99.85 mass %. The results of chemical analysis for Cd, Co, Ni, and Zn concentration were almost same as the values for starting sample. Cu and Mn concentration was decreased during the process in the range from 11.7 mass % to 27.4 mass %, and in the range from 11 mass % to 20.7 mass % for Mn and Cu, respectively.

## 3.2. Adsorption Method

The above-mentioned results of the two-step neutralization treatment revealed that most of Fe (ferric iron) and Cu were separated and removed as the precipitates from the real AMD water derived from the Robule Lake at pH 4 and 7, respectively. However, approx. 30 mg L$^{-1}$ of Mn till remained in the treated AMD water. Generally, the Mn$^{2+}$ ions removal from water can be induced by the neutralization treatment at pH 9 or above, which is often more than the pH value for the effluent standard. In this case, it is necessary for discharge to the river or surrounding areas to lower pH to below the effluent standard. Thus, the adsorption method using chitosan, a kind of polysaccharide, was investigated as a new approach for Mn$^{2+}$ ion removal.

To confirm the Mn$^{2+}$ ion removal capacity of chitosan, the adsorption removal tests were firstly carried out using the simulated Mn solutions (0.5 mM). Mn$^{2+}$ ion removal from the aqueous solution at a higher pH (more than 6.5) was identified, while Mn removal capacity by chitosan was not confirmed below pH 5. Figure 5 shows change in the Mn removal rate when various input amounts of chitosan (0.5, 1, 2, 4 g) were added to 50 mL of the Mn solutions with pH 6.7 and 7. In case of 0.5 g of chitosan input, approx. 30 mass % of Mn removal rate was indicated. One gram of chitosan input increased the removal rate to 50 mass %. Finally, Mn removal rate reached 90 mass % when 4 g of chitosan was applied to the adsorbent for 0.5 mM of Mn solution.

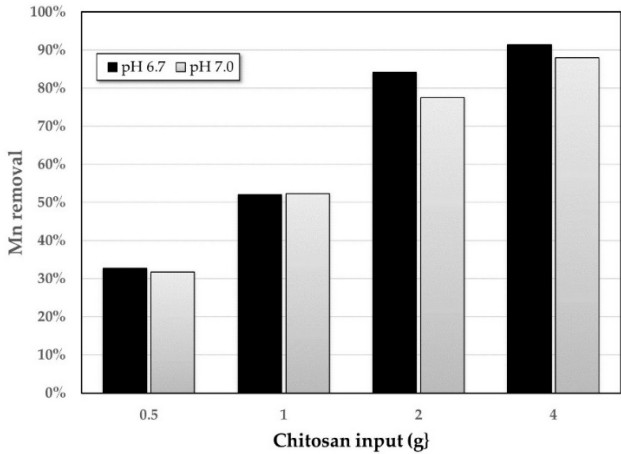

**Figure 5.** Effect of input amounts of chitosan on Mn removal from the simulated solutions at pH 6.7 and 7.

Figure 5 shows change in $Mn^{2+}$ ion concentration and the amount of Mn removal as a function of input amount of chitosan. Due to the adsorption removal using chitosan, the Mn concentration in solutions reduced from 25 mg $L^{-1}$ (0.5 mM) to approx. 3 mg $L^{-1}$ with an increase in chitosan input. Chitosan input increased the solution pH to around 8. Chitosan, a cationic polymer obtained from chitin, has a lot of amino groups (-$NH_2$). Increase in the solution pH may be caused by the consumption of H+ due to transferring -$NH_2$ to -$NH^{3+}$. Consequently, these results revealed that chitosan had a good performance for Mn removal from the simulated solution. Then, the application of chitosan to Mn removal from the real AMD water was attempted (Figure 6).

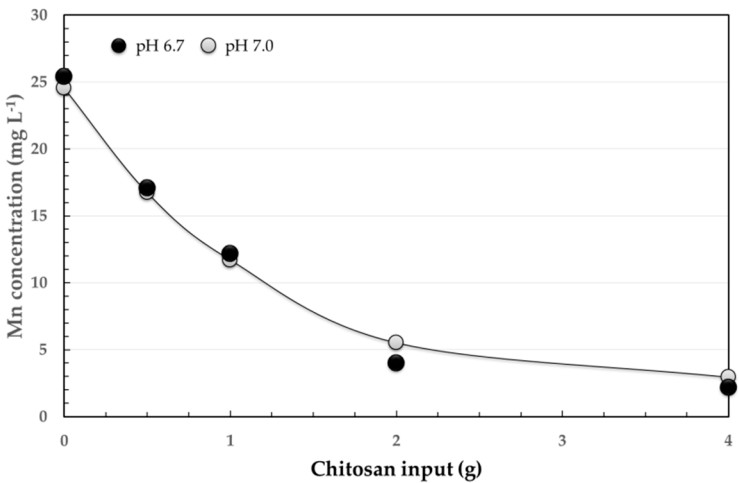

**Figure 6.** Change in Mn concentration and the amount of Mn removal at pH 6.7 and 7 as a function of input amount of chitosan.

In this experiment, the real AMD water sample, which was derived from the Robule Lake, was used. From this AMD sample, almost all of the Fe and Cu were already removed due to the two-step neutralization treatment at pH 4 and 7, respectively. The treated AMD sample had the pH value of 7.2, and finally contained about 35 mg $L^{-1}$ of Mn, 427 mg $L^{-1}$ of Ca and 1650 mg $L^{-1}$ of sulfur. Then, the Mn removal test with 5 g of chitosan as an adsorbent was conducted. Figure 7 shows change in the Mn concentration before and after the adsorption removal test using 5 g of chitosan. The results of Mn concentration in the simulated solutions are also presented in Figure 7. Due to the adsorption into chitosan, the Mn concentration in the simulated solutions reduced to around 2 mg $L^{-1}$. This result was equivalent to approx. 90 mass % of Mn removal. Investigations are continuing by incubating the chitosan in the treated AMD water for 24 h. As a result, the Mn concentration decreased to about

10 mg L$^{-1}$, higher than that in the simulated solutions (Figure 7). It was considered that this result might be caused by the co-existing elements such as Ca, S, etc. Actually, 44 mass % of Ca and 50 mass % of S were also removed from the treated AMD. Sulfur content is mainly derived from sulfate ion (SO$_4^{2-}$). It was confirmed that chitosan contributed to the removal of not only Mn but SO4$^{2-}$ from the treated AMD at below pH 8. Chitosan derives from external skin of Crustacea such as crab shell which corresponds to an inedible part. It is suggested that wastes from food processing industries for marine products, etc. can be used for chitosan. However, the removal efficiency leaves much to be desired. In this study, the adsorption removal capacity was mainly investigated. Some approaches will be required to advance the efficiency and apply to the actual operation. For that, it is considered that a deacetylation degree of chitosan will be improved, and that chitosan hydrogels will be prepared with some other materials which have the higher adsorption capacity, and so on.

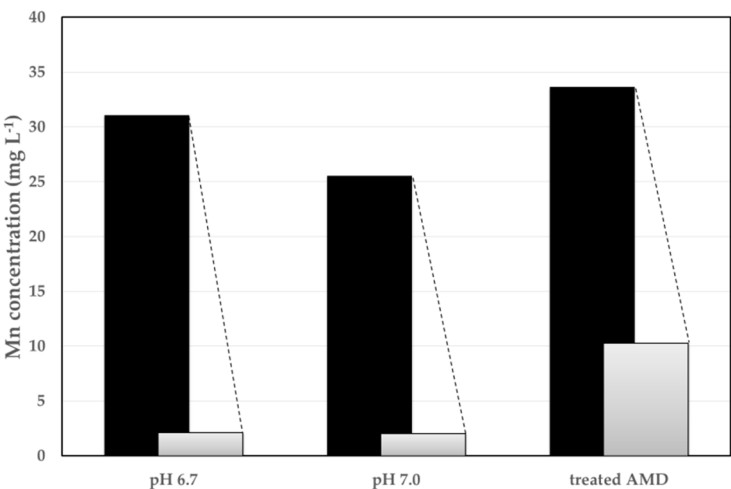

**Figure 7.** Change in Mn concentration before and after the adsorption removal test using 5 g of chitosan.

### 3.3. Characterization of the Obtained Sludge

This study was carried out to purify the AMD generated in the area of copper mining activities in the southeast of Serbia. It is also taking the recycling of metals in the AMD into consideration. In this case, Fe and Cu are main components in the AMD. The possibility for re-use of these metals was investigated. From the results of the laboratory test and semi-industrial plant operation, the two-step neutralization process through pH 4 and 7 could remove Fe, Cu, Zn, Co, Ni, and Cd from the AMD water effectively. It was also confirmed Fe and Cu could mainly be recovered at pH 4 and 7, respectively. Table 6 shows a chemical composition of the obtained sludge due to the two-step neutralization process.

**Table 6.** Chemical composition of the sludge obtained in the two-step neutralization process.

| Element Mass % | First Neutralization Step (pH Initial–pH 4) | Second Neutralization Step (pH 4–pH 7) |
|---|---|---|
| Al | 2.73 | 7.13 |
| Cu | 0.15 | 1.24 |
| Fe | 33.52 | 0.47 |
| Ca | 3.07 | 11.77 |
| Mg | 0.015 | 2.79 |
| Mn | 0.26 | 1.67 |
| Zn | 0.013 | 0.50 |
| As | 0.0024 | 0.0014 |
| Cd | 0.000018 | 0.00174 |
| Co | 0.000885 | 0.0374 |
| Ni | 0.000651 | 0.0189 |

During the second neutralization step, elements concentration reducing have the tendency: Cu > Zn > Co > Ni > Cd > Mn. Manganese concentration has a higher value than maximum allowed value according to the Serbian legislation [24].

The results of the sludge chemical characterization are presented in Table 6.

The obtained sludge at pH 4 (1st neutralization step) contained 33.5 mass % of Fe (Table 6). This result indicates that this sludge was mainly iron hydroxide. This supports Fe concentration in the real AMD sample drastically decreased up to pH 4. Especially, the obtained sludge at pH 7 (2nd neutralization step) mainly contained 1.2 mass % of Cu (Table 6). This Cu grade of sludge is higher than that of the copper ore which has been currently mined. It is expected that the obtained sludge including Cu will be applied as a raw material for Cu metal through some operations such as the acid leaching, solvent extraction and electrochemical methods, and applied in the existing Copper Smelter in Company Serbia Zijin Bor Copper. Based on the requirements of the steel processing plants, sludge obtained on pH 4 will be direct used or used after the appropriate treatment.

Consequently, it can be assumed that a combination process including neutralization and adsorption methods is applied to the AMD generated from some copper mines in the southeast Serbia. Figure 8 shows a schematic illustration of the assumed AMD treatment process for metal removal and recovery.

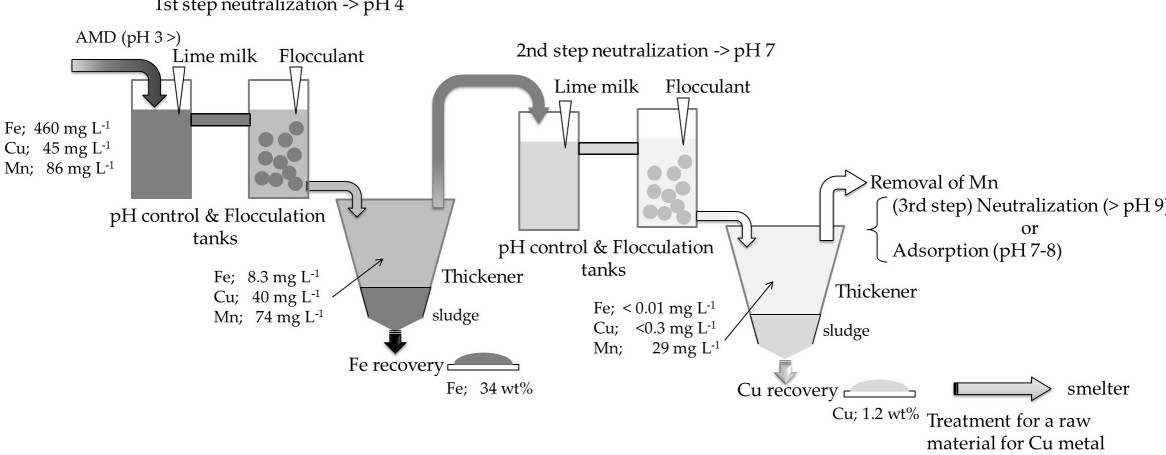

**Figure 8.** Schematic illustration of the assumed AMD treatment process for metal removal and recovery.

The two-step neutralization process with pH 4 and 7 is applied to the real AMD with pH value of less than 3. Through this operation, Fe and Cu are removed and recovered separately. For instance, it is suggested that recovered Fe sludge is available for a raw material for an inorganic flocculant such as ferric polysulfate. The recovered Cu sludge through some treatments will finally contribute to Cu metal production. The treated AMD still contains most of the $Mn^{2+}$ ions which has a higher value than the maximum allowed value according to the Serbian legislation. In this case, there are two options for Mn removal. One option is Application of the 3rd step neutralization to the treated AMD. However, the pH of treated wastewater should be reduced again below the effluent standard. It is also expected that adsorption method using chitosan is applied for Mn removal. This method will be operated below the pH value for the effluent standard. Some improvements nay be required for the application of adsorption method to the actual removal of Mn. It was indicated that the combination process with neutralization and adsorption methods would have the possibility for an effective purification to the real AMD water and recycling of some metals.

## 4. Conclusions

In this study, the possibility for a new approach of metal ion removal was investigated. This approach included the two-step neutralization and adsorption methods and was applied to the

real AMD water generated from the copper mines in the southeast of Serbia. As for the neutralization process, the experiment results revealed that the two-step pH control neutralization and precipitation method was effective and recovered Fe and Cu separately in the sludge generated along the process. It was confirmed that the AMD water quality was drastically improved. Manganese ions still remained in the treated AMD water after the two-step neutralization. Thus, the Mn removal using natural organic polymer as an adsorbent was investigated. In the adsorption removal tests using simulated solutions, Mn removal from the aqueous solution at a higher pH (more than 6.5) was identified. Mn removal rate reached 90 mass % when 4 g of chitosan was applied to the adsorbent for 0.5 mM of Mn solution. It was also indicated that 5 g of chitosan removed 70 mass % of Mn from 50 mL of the treated AMD at below pH 8. Besides, the obtained sludge at pH 7 contained 1.2 mass % of copper with some amounts of the other elements. Thus, it is expected that the combination process with the two-step neutralization and adsorption methods will contribute to not only the AMD purification but also to Cu recycling. This investigation will contribute to the sustainable development for mining operations in near future.

**Author Contributions:** R.M. and M.B. Conceptualization and methodology; R.M. and M.B. wrote the first draft; D.B., T.A.T., V.G. data acquisition and data interpretation, N.M. and Z.S. supervision., R.M. and M.B. writing review. All authors have read and agreed to the published version of the manuscript.

**Funding:** This paper is result of Japan, Serbia mutual JICA Project in the SATREPS program: "The Project for the Research on the Integration System for Spatial Environment Analysis", funded from the donation of the Japanese Government, with the participation of Serbian funding and Project No. 37001, "The Impact of Mining Waste from RTB Bor on the Pollution of Surrounding Water Systems with the Proposal of Measures and Procedures for Reduction the Harmful Effects on Environment", funded by the Ministry of Education and Science in the Republic of Serbia.

**Acknowledgments:** The research presented in this paper was done with support of the Ministry of Education, Science and Technological Development of the Republic of Serbia, within the scientific research work at the Mining and Metallurgy Institute Bor, according to the contract with registration number 451–03-68/2020–14/200052.

**Conflicts of Interest:** The authors declare no conflict of interest.

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
