# Peer review of "New Approach of Metals Removal from Acid Mine Drainage"

_applsci, doi:10.3390/app10175925_

Round 1
Reviewer 1 Report
This is a good manuscript containing sufficient data and analysis, and it is especially practically meaningful for mining wastewater treatment. It provides valuable data and information for the mining industry, especially the treatment of acid mine drainage. Before publication, there are some minor issues that need improvement.
- line 25, this sentence needs re-written.
- line 43, be careful about the abbreviation. What does DG stand for here?
- line 113, delete "are".
- lines 124 to 126, this sentence needs re-written. Also, delete "And" in the line 126.
- line 127, please specify which Mn ion, Mn2+?
- line 165, there are two punctuations.
- lines 244 to 252, this information is about the water characteristics and it should go to section 2, beside table 1.
- No error bar has been found in any table or figure. How do the authors confirm the reproducibility? Were the tests only done one time?
Reviewer 2 Report
The paper is interesting but in my opinion it lacks novelty.
Precipitation and sorption are classic methods for metal removal from water.
Thus, the authors must state clearly in the manuscript what are the main contributions of their work.
Additionally, a thorough revision of the English (grammar, sintaxis...) is necessary.
Reviewer 3 Report
This manuscript describes the integrated treatment of Acid mine drainage effluents (combining selective precipitation steps and sorption on chitosan) on real case (effluents from Serbian copper mine). This topic is worth of interest, though the novelty is debatable (however, it is always interesting enriching the literature on the treatment of real cases, using pilot-scale units).The introduction finely justifies the strategy though the appropriateness of using chitosan as a “cheap” sorbent for the treatment of high flow rates remains debatable (for Mn removal).
Editing
English editing (some style and grammar or many typing mistakes) should be improved; though the text is globally correct and readable.
The editing quality of some figures should be improved (sharpness of Figure 5, for example, the choice of the y-axis label and 100%, 90%, 80%; since the title of the axis already includes the % unit, mentioning again on the tags is probably not necessary). Figures 6-8 could be also improved for sharpness.
Specific comments
Chitosan characteristics are important to be reported (molecular weight, acetylation degree). Indeed, the use of biological-based sorbents may be facing problems of reproducibility due to the intrinsic variability of the resource.
Using chitosan as powder or chips means that the authors are processing in batch system. Is it the suggested method for pilot-scale unit? It is a pity the authors did not consider the sorption step in their global pilot-scale unit. Usually sorbents are preferentially used in column systems (like resins); however, chitosan powder considerably limits the possibility to operate large flow rates due to clogging and hydrodynamic problems. This should be discussed.
Characterizing the precipitate with, for example, EDX semi-quantitative analysis would readily bring information on the composition of the solid phase and the selectivity of the precipitation steps (in complement to the ICP analysis of treated water). In section 3.3. (Characterization of obtained sludge at pH 7 for Cu resource; the title of this section could be changed because the authors also included some information on pH 4 precipitation step), the authors discuss the composition of the sludge but apparently the method used for acquiring these data is not documented (or I missed it). This information, in turn, would help in discussing the possibility to valorize these precipitates. Indeed, the authors did not sufficient discuss how the precipitates could be processed at the end of the treatment. Processing of precipitates for metal valorization or storage? What could be the stability of the precipitates for storage (including metal release from a solid phase that may contain high concentration of hazardous compounds)?
Manganese treatment by chitosan. Figure 6 clearly demonstrates that the sorption capacity is dramatically weak (less than 1 mg Mn/g). Does it make sense for a practical and competitive application?
Processing the treatment of significant amounts of AMD is interesting. Probably, completing the Figure 8 with data on composition of solid phases, liquid phases (perhaps as additional material for avoiding overloading the manuscript) would bring a fine overview of the process. I wonder if providing a kind of cost analysis would not bring a significant insight of interest for the readers of this paper. Enriching the paper with this economical statement would probably increase the impact of the work. There is an increasing demand for stating the economic implications; this is not easy but readers would appreciate (this is at least my own case).
Author Response
Response to Reviewer 3 Comments
English editing (some style and grammar or many typing mistakes) should be improved; though the text is globally correct and readable.
Thorough revision of the English is made by the licensed translator.
The editing quality of some figures should be improved (sharpness of Figure 5, for example, the choice of the y-axis label and 100%, 90%, 80%; since the title of the axis already includes the % unit, mentioning again on the tags is probably not necessary). Figures 6-8 could be also improved for sharpness.
Figure 5-8 are revised.
Specific comments
Chitosan characteristics are important to be reported (molecular weight, acetylation degree). Indeed, the use of biological-based sorbents may be facing problems of reproducibility due to the intrinsic variability of the resource.
Using chitosan as powder or chips means that the authors are processing in batch system. Is it the suggested method for pilot-scale unit? It is a pity the authors did not consider the sorption step in their global pilot-scale unit. Usually sorbents are preferentially used in column systems (like resins); however, chitosan powder considerably limits the possibility to operate large flow rates due to clogging and hydrodynamic problems. This should be discussed.
Lines 177-181: As an adsorbent material for metal removal, chitosan with medium molecular weight was purchased from Sigma-Aldrich Japan Co., Japan. This chitosan sample had a deacetylation degree of 75 to 85% and 1 wt % of the chitosan in 1% acetic acid at 25 °C had 200-800 cps of viscosity. The chitosan sample having a shape of powder and/or chips was used without any purification for the adsorption removal test.
Lines 391-401 – It was confirmed that chitosan contributed to removal of not only Mn and but SO42- from the treated AMD at below pH 8. Chitosan derives from external skin of Crustacea such as crab shell which corresponds to an inedible part. It is suggested that wastes from food processing industries for marine products, etc. can be used for chitosan. However, the removal efficiency leaves to be desired. In this study, the adsorption removal capacity was mainly investigated. Some approaches will be required to advance the efficiency and apply to the actual operation. For that, it is considered that a deacetylation degree of chitosan will be improved, and that chitosan hydrogels will be prepared with some other materials which have the higher adsorption capacity, and so on.
Characterizing the precipitate with, for example, EDX semi-quantitative analysis would readily bring information on the composition of the solid phase and the selectivity of the precipitation steps (in complement to the ICP analysis of treated water). In section 3.3. (Characterization of obtained sludge at pH 7 for Cu resource; the title of this section could be changed because the authors also included some information on pH 4 precipitation step), the authors discuss the composition of the sludge but apparently the method used for acquiring these data is not documented (or I missed it). This information, in turn, would help in discussing the possibility to valorize these precipitates. Indeed, the authors did not sufficient discuss how the precipitates could be processed at the end of the treatment. Processing of precipitates for metal valorization or storage? What could be the stability of the precipitates for storage (including metal release from a solid phase that may contain high concentration of hazardous compounds)?
Title of the 3.3. section is revised according to Reviewer comment.
Lines 407-410: This study was carried out to purify the AMD generated in the area of copper mining activities in the southeast Serbia. It is also taking the recycling of metals in the AMD into consideration. In this case, Fe and Cu are main components in the AMD. The possibility for re-use of these metals was investigated.
Lines 427-429 – revised text for precipitated processing: … and applied in the existing Copper Smelter in Company Serbia Zijin Bor Copper. Based on the requirements of the steel processing plants, sludge obtained on pH 4 will be direct used or used after the appropriate treatment.
In a part 2.4 is given technique for sludge chemical characterization.
Manganese treatment by chitosan. Figure 6 clearly demonstrates that the sorption capacity is dramatically weak (less than 1 mg Mn/g). Does it make sense for a practical and competitive application?
Processing the treatment of significant amounts of AMD is interesting. Probably, completing the Figure 8 with data on composition of solid phases, liquid phases (perhaps as additional material for avoiding overloading the manuscript) would bring a fine overview of the process. I wonder if providing a kind of cost analysis would not bring a significant insight of interest for the readers of this paper. Enriching the paper with this economical statement would probably increase the impact of the work. There is an increasing demand for stating the economic implications; this is not easy but readers would appreciate (this is at least my own case).
Figure 6 is revised and the Figure 8 is completing with data on composition of solid phases and liquid phases.
Line 438-450 - This paragraph is revised for Reviewer 2 & 3 comments: For instance, it is suggested that recovered Fe sludge is available for a raw material for an inorganic flocculant such as ferric polysulfate. The recovered Cu sludge through some treatments will finally contribute to Cu metal production. The treated AMD still contains most of Mn2+ ions which has the higher value than maximum allowed value according to the Serbian legislation. In this case, there are two options for Mn removal. One option is Application of the 3rd step neutralization to the treated AMD. But, the pH of treated wastewater should be reduced again below the effluent standard. It is also expected that adsorption method using chitosan is applied for Mn removal. This method will be operated below the pH value for the effluent standard. Some improvements nay be required for the application of adsorption method to the actual removal of Mn. It was indicated that the combination process with neutralization and adsorption methods would have the possibility for an effective purification to the real AMD water and recycling of some metals.
Line 467 – 468 - This investigation will contribute to the sustainable development for mining operation in near future.

Round 2
Reviewer 2 Report
The paper has been sufficiently improved to be accepted.
Reviewer 3 Report
I consider the authors made sufficient efforts in the revision of the manuscript for making the paper publishable.